# Exploring the Influence of Morphology on Bipolaron–Polaron Ratios and Conductivity in Polypyrrole in the Presence of Surfactants

**DOI:** 10.3390/molecules29061197

**Published:** 2024-03-07

**Authors:** Thaneeya Samwang, Nozomi Morishita Watanabe, Yukihiro Okamoto, Hiroshi Umakoshi

**Affiliations:** 1Division of Chemical Engineering, Graduate School of Engineering Science, Osaka University, 1-3 Machikaneyamacho, Toyonaka 560-8531, Osaka, Japan; no.watanabe.es@osaka-u.ac.jp (N.M.W.); okamoto.yukihiro.es@osaka-u.ac.jp (Y.O.); 2Department of Chemical Engineering, Faculty of Engineering, Mahidol University, 25/25 Phuttamonthon 4 Road, Salaya, Phuttamonthon, Nakhon Pathom 73170, Thailand

**Keywords:** polypyrrole, surfactant, sodium dodecyl sulfate, bipolaron and polaron ratio, C-H deformation, Raman spectra

## Abstract

This research aims to deepen the understanding of the relationship between conductivity and morphology in polypyrrole (PPy) via a comparison of the bipolaron to polaron ratios with a focus on the C-H deformation area. PPy samples were synthesized with different surfactants: sodium dodecyl sulfate (SDS), cetyltrimethylammonium bromide (CTAB), and tween 80 (TW). This study revealed that SDS significantly altered the bipolaron and polaron in the C-H deformation region and showed higher conductivity than other surfactants. Notably, the morphological shifts to a sheet-like structure when using ammonium sulfate (APS) contrasted with the particle-like form observed with ferric chloride (FeCl_3_). These results showed that if the oxidant changed, the bipolaron and polaron ratios in C-H deformation were unrelated to PPy morphology. However, this work showed a consistent relationship between SDS use, the bipolaron and polaron ratios in the C-H deformation, and the conductivity properties. Moreover, the natural positive charge of PPy and negatively charged SDS molecules may lead to an electrostatic interaction between PPy and SDS. This work assumes that this interaction might cause the transformation of polaron to bipolaron in the C–H deformation region, resulting in improved conductivity of PPy. This work offers more support for the future investigation of PPy characteristics.

## 1. Introduction

In recent times, conductive polymers, specifically polypyrrole (PPy), have garnered significant interest in academic and industry research communities globally due to their distinctive characteristics and promise for diverse commercial applications. The increased interest in PPy originates mostly from its remarkable thermal and environmental durability, impressive electrical conductivity, and simple synthesis procedure [1,2,3]. These polymers are conductive because they have a conjugated electron system composed of alternating single and double bonds within their chemical structure [4]. The unique chemical structure of conductive polymers not only allows them to conduct electricity but also piques the interest of researchers in other fields, leading to breakthroughs in conductive polymers and applications such as biomaterials [5,6], biosensors [7,8,9,10], and energy storage [11,12].

The electrical properties of PPy were also modified based on the polymerization process. The chemical polymerization technique is a standard method for producing PPy particles. This method required oxidizing agents, such as ammonium persulfate (APS) or ferric chloride (FeCl_3_), to oxidize the pyrrole monomer (Py), which initiated the polymerization process. It is theorized that during the first phase of pyrrole oxidation, insoluble pyrrole oligomers form, and then PPy chains arise from clustered aggregates, culminating in globular forms. [13,14,15,16]. The morphology of PPy also was found to alter from globular structures to another form using templates, such as hard templates and soft templates. When utilizing a hard template, the final product’s morphology is significantly influenced by the pre-prepared material, such as anodic alumina membrane producing nanowire morphology of PPy [17], silicon wafer [18], or polymer [19]. Following the reaction, the template is then carefully removed. The utilization of soft templates in polymerization has several notable benefits, particularly regarding the flexibility and ease it provides to the polymer structure. Soft templates, typically composed of organic compounds or biological entities, allow for a more controlled and flexible synthesis process [20]. The inherent flexibility of soft templates accommodates a broader range of polymer sizes and shapes, catering to specific application needs, such as surfactants like fatty alcohol polyoxyethylene ether surfactants [21], sodium dodecyl benzene sulfonate (SDBS) [22], or sodium dodecyl sulfate (SDS) [23,24]. Additionally, these surfactants, when utilized as soft templates in polymer synthesis, provide a unique and efficient method for creating polymers with the appropriate structural and functional features. In solution, these amphiphilic compounds spontaneously form micelles or other self-assembled structures, producing nanoscale templates for monomers to polymerize polymers with various morphologies and sizes [25,26,27,28]. Moreover, using an anionic surfactant, such as SDS or SDBS, can enhance the conductivity of PPy [3,29]. Meanwhile, many studies reported that the non-spherical structure of PPy obtained using CTAB showed higher electrical properties than the spherical shape [30]. Additionally, Khadem et al. [31] reported that using CTAB to prepare PPy could give it a string-bead shape with better conductivity than a spherical shape made from SDS in the same conditions. Yussuf et al. [32] found PPy prepared in SDS with ferric chloride (FeCl_3_) as an oxidizing agent to have fibrillar morphology and show slightly higher conductivity than the other oxidizing agent, APS.

The surfactant enhances the electronic properties of PPy through a process called doping, which often involves oxidation occurring during polymerization. The oxidation level of PPy is intricately linked with the formation of polarons and bipolarons, which are localized defects causing lattice distortion within the polymer’s structure (Figure 1a). [33,34] These defects are crucial for facilitating the mobility of charge carriers, where polarons and bipolarons correspond to low and high oxidative states, respectively [35,36]. The genesis of these polarons and bipolarons is highly dependent on the synthesis conditions and plays a significant role in the chemical and electrical properties of PPy. Research conducted by Santos et al. [37] and Pang et al. [38] through Raman spectroscopy analysis on the chemical polymerization of PPy highlighted that the pyrrole rings predominantly attach at the alpha–alpha (α–α) positions, though they can also link at the beta–beta (β–β) and beta–alpha (β–α) positions (Figure 1b). This variation in binding sites influences the molecular configuration of dimers and trimers in PPy, as well as their respective oxidative states, manifested as bipolarons and polarons. Further research by Ishpal and Kaur [39], Trchová et al. [40], Tumacder et al. [41], and Paúrová et al. [42] showed the different Raman spectral profiles of bipolarons and polarons of PPy having varying nanostructures. Building on previous research [24], it was observed that PPy polymerized with SDS presented a sheet-like morphology and demonstrated an increased bipolaron/polaron ratio in line with Py concentration, which correlates with a decrease in the bandgap. This contrasts with PPy synthesized without SDS, which typically forms spherical structures.

Therefore, this study delves into the interplay between conductivity and morphology in PPy by examining the bipolaron and polaron ratios, particularly in the C–H deformation region of the PPy structure from the Raman spectra. We analyzed the bipolaron and polaron ratio in PPy polymerized with different surfactants, including SDS, CTAB, and Tween 80 (TW). Our theory is that PPy made with CTAB and TW will have a spherical shape and a lower bipolaron/polaron ratio, especially in the C–H deformation region, compared to PPy that was treated with SDS. According to the underlying idea, a low bipolaron/polaron ratio in the C–H deformation region of spherical PPy structures might mean that they conduct electricity less well. Through this investigation, we aim to clarify the gap reason in the relationship between the morphology and electrical properties of PPy through bipolaron and polaron ratios (Figure 1).

## 2. Results

This section presents the findings of the experiment, encompassing the physical, chemical, and electrical characteristics of the PPy samples synthesized by the chemical polymerization method. The PPy samples consisted of Py polymerized with different surfactants: SDS, CTAB, or TW. Additionally, two types of oxidizing agents, APS and FeCl_3_, were used. The samples were labeled Py-SDS-APS, Py-CTAB-APS, Py-TW-APS, Py-SDS-Fe, and Py-CTAB-Fe compared with Py polymerization without the surfactant (PPy*), as shown in Table 1. The concentrations of each component—Py monomer, surfactants, and oxidizing agents—utilized in this experiment were constant (0.10 M), where the concentration of surfactants exceeded the critical micelle concentration (CMC), allowing them to create a micelle template in the water system.

### 2.1. Average Size and Zeta Potential

The average size of six PPy samples, each generated with different surfactants and oxidizing agents, is shown in Figure 2a. The results highlight the significant impact of the polymerization process on the hydrodynamic diameter (particle size), as determined by the Dynamic Light Scattering (DLS) technique using the zetasizer apparatus. Notably, PPy synthesized with APS as the oxidizing agent and surfactant exhibited a smaller size of approximately 100–400 nm compared to PPy synthesized with FeCl_3_ as the oxidizing agent (800–1200 nm) or without the surfactant (approximately 2500 nm). Py-SDS-APS had the smallest size around 100 nm, followed by Py-TW-APS, Py-CTAB-APS, Py-SDS-Fe, and Py-CTAB-Fe. PPy* exhibited the largest size as measured by DLS. Additionally, the polydispersity index (PdI) was used to indicate the uniformity or heterogeneity of particle size distributions, as shown in the same figure (Figure 2a). Py-TW-APS and Py-SDS-APS showed PdI values of 0.12 and 0.19, respectively. Py-CTAB-APS, Py-SDS-Fe, and PPy* had a greater polydispersity than 0.30 (0.40, 0.42, and 0.63, respectively), with Py-CTAB-Fe having the highest PdI of 0.86. A low PdI indicates a narrower size distribution, suggesting a relatively uniform particle size, whereas a higher PdI indicates a broader size distribution with more variability [43].

Figure 2b shows the zeta potential. The zeta potential is a measure of the stability of particles in a water-based solution. Ultrapure water was used as the base solution for all preparatory studies. The zeta potentials of PPy varied based on the type of surfactant used. PPy polymerized with an anionic surfactant (SDS) had a negatively charged surface; Py-SDS-APS is −24.80 mV and Py-SDS-Fe is −25.70 mV. For CTAB, the cationic surfactants Py-CTAB-APS and Py-CTAB-Fe had positively charged surfaces of 22.30 and 41.10 mV, respectively. Despite the fact that TW is a nonionic surfactant, Py-TW-APS has a positive charge of 16.10 mV since PPy* is a natural positive charge polymer (26.60 mV). As the concentration of TW increased from 2 to 5 to 10 wt%, the zeta potential of PPy decreased slightly from 17.20 to 16.10 to 15.50 mV, respectively, suggesting that it approached zero. 

### 2.2. Morphology and Structure Characteristics

The morphology of the PPy samples was examined by scanning electron microscopy (SEM), as shown in Figure 3a–f, which show the SEM images at a magnification of 10,000. The original pictures from SEM were compared to the analyzed pictures using Image J software (Version 1.53 K) to evaluate the edges and shape of particles (evaluated using the function Find Edges), as shown on the right of each sample. 

PPy polymerized without a surfactant (PPy*) had a sphere-like shape, whereas Py-SDS-Fe and Py-CTAB-Fe exhibited particle clusters. A string of sphere-like forms was observed in the Py-CTAB-APS sample, and a sheet-like structure was created using SDS as the surfactant in Py-SDS-APS. The images of the Py-TW-APS sample (Figure 3d) were not clear because they could not be completely dried using a freeze-drying process. This may be because Tween 80 has a low melting point (−25 °C) (referred to in the Safety Data Sheet of Tween 80 (CAS number 9005-65-6) from TCI America, revision number 2), so it maintains the structure of PPy in a semi-liquid state at room temperature and is difficult to observe by SEM. However, after using Image J software to analyze it, it was determined to be a particle-like form because we found the edge in a round shape. The various morphologies of PPy examined by SEM proved that the surfactants and oxidizing agents used influenced the PPy shape.

### 2.3. Electrical Conductivity

The resistance of the dry samples was determined using a chemiresistance apparatus, as described in the Materials and Techniques section, to define the electrical properties of the PPy samples. The resistance (R) is derived from the reverse value of the slope from the graph correlation between the voltage (V) and current (I) using Ohm’s law (V = IR and R = 1/slope), as shown in Figure 4a. Subsequently, the conductivity of the samples was calculated from the resistance using the equation described in the Materials and Methods section, as shown in Figure 4b. The results showed that PPy samples prepared with SDS had higher conductivity, arranged in the following order: Py-SDS-Fe, Py-SDS-APS, Py-CTAB-APS, Py-CTAB-Fe, and PPy*. However, the conductivity of PPy* was not significantly different from that of Py-CTAB–Fe (*p* > 0.05). For the Py-TW-APS sample, the electrical conductivity was not measurable because it could not be dried using a freeze-drying process, as mentioned in Section 2.2. 

### 2.4. Raman Spectra and Bipolarons/Polarons Ratio

After dialysis, the Raman spectra of PPy suspensions were analyzed using Raman spectroscopy with a 532 nm excitation laser to determine the chemical information of PPy, with a focus on peak characterization, and to calculate the bipolaron and polaron ratio. This study focused on three regions: C–C ring deformation, C–H deformation, and C–C ring stretching. In each region, both the bipolaron and polaron structures can be found in different Raman shift wavenumbers depending on the PPy production. For C–C ring deformation, bipolarons were observed at 926–939 cm^−1^, and polarons were observed at 963–985 cm^−1^. The C–H deformation region exhibited polarons at 1050–1056 cm^−1^ and bipolarons at 1079–1084 cm^−1^. C–C ring stretching was observed for polarons at 1330–1344 cm^−1^ and for bipolarons at 1377–1382 cm^−1^. The polarons associated with the C=C stretching peak were observed within the range of 1562–1593 cm^−1,^ and the presence of the bipolaron structure was observed within the range of 1600–1618 cm^−1^ [37,40,41,44,45].

Figure 5a shows the Raman spectra of the PPy samples. In the C–C ring deformation region, the peak bipolaron of PPy* is at 931 cm^−1^, and it is obvious that the bipolaron of Py-SDS-APS shifted to the right (from 931 to 945 cm^−1^) and that of Py-CTAB-APS and Py-CTAB-Fe shifted to the left (to 922 cm^−1^). While the polaron of the C–C ring deformation of PPy* is at 967 cm^−1^, from this point, Py-SDS-APS shows a slight shift from 967 cm^−1^ to 973 cm^−1^. In addition, those of Py-CTAB-APS and Py-CTAB-Fe shifted to the right to 987 cm^−1^. Notably, the spectral positions of the bipolaron and polaron in the Py-CTAB-APS and Py-CTAB-Fe samples clearly shifted out from each other compared to the other samples. The shifted peaks of polarons far from bipolarons indicated that Py-CTAB-Fe has a lower oxidative chain of PPy [39]. For the C–H deformation area, we found that in Py-TW-APS and Py-CTAB-Fe, there was no peak in the bipolaron structure compared to the other samples. For Py-SDS-APS and Py-SDS-Fe, the polaron shifted from reference line 1048 to the right, indicating an increase in oxidized PPy [46]. For C–C ring stretching, all samples exhibited a polaron structure at 1333–1342 cm^−1^ and a bipolaron at 1376–1379 cm^−1^, whereas Py-CTAB-Fe had no bipolaron peak but showed a peak shift at 1411 cm^−1^, which was attributed to the antisymmetrical inter-ring C–N stretching of oxidized PPy [34,44,47]. The polarons associated with the C=C stretching peak were observed in every sample in the range of 1563–1583 cm^−1^, but the bipolaron structure connected to the C=C stretching cannot be determined. However, the C=C stretching area exhibited the most prominent peak feature observed when comparing each sample. A broader morphology of the peak can be discerned in the Py-CTAB-Fe and Py-CTAB-APS samples, whereas it appeared narrower in the Py-TW-APS, Py-SDS-Fe, and Py-SDS-APS samples.

Figure 5b illustrates the ratios of bipolaron and polaron peak intensities in relation to other locations, namely C–C ring deformation, C–H deformation, and C–C ring stretching, except in the C=C stretching region because the sample produced for this study could not be discovered; as a result, the ratio in this particular area remains undetermined. The apparent variation in the ratio appears in the C–H deformation region; the bipolaron and polaron ratios of PPy produced using SDS were greater than those of the other samples, even when either APS or FeCl_3_ was used as an oxidizing agent. These ratios were used to further investigate the bipolaron and polaron structures in the PPy samples, which may provide information on the electronic properties of PPy.

### 2.5. UV-Vis-NIR Absorption

Figure 6a shows the UV-Vis-NIR absorption spectra. The spectral characteristics were slightly influenced by the presence of the surfactant and oxidizing agent and supported the oxidative structure of the samples found in the Raman spectra. All PPy samples presented their maximum absorption around 460 nm in ultrapure water, which corresponds to π-π* transition and refers to the neutral and polarons in PPy [4,40,48]. In addition, the samples represented a peak over the wavelength of 800 nm and up to 970 nm, and the peak around 800–900 nm referred to the oxidized bipolarons of PPy [4,49]. Figure 6b shows the bandgap calculated from Tauc’s plot as described in the Materials and Methods section. These data were used to support the conductivity of PPy samples. The low bandgap referred to high conductivity properties due to the reduction in the gap between the valence band and the conduction band [50,51].

## 3. Discussion

The DLS experiment showed that Py-SDS-APS presented the smallest diameter, followed by Py-TW-APS, Py-CTAB-APS, Py-SDS-Fe, Py-CTAB-Fe, and PPy*. Notably, using a surfactant can reduce the PPy size compared to not using a surfactant. In addition, polymers with anionic surfactants, such as SDS, tended to have a smaller average size than those made with cationic surfactants, such as CTAB, or nonionic surfactants, such as TW, which was similarly found in Hoshina Y. et al., Paurova M. et al., and Zoromba M. Sh. et al., for example [14,42,52]. This could be because the negatively charged SDS (OSO_3_^−^ groups) can stabilize the growing PPy particles, thereby preventing excessive growth and aggregation [51]. When using constant SDS or CTAB, the size differed depending on the type of oxidizing agent used. When APS was utilized, the PPy size was lower than that of FeCl_3_, as shown in Figure 2a, which is similar to the hydrogel created by Bo et al., even though the unreacted molecules did not wash away [53]. In addition, FeCl_3_ may substantially increase the ionic strength of the solution compared to APS, and higher ionic strengths can diminish electrostatic repulsion among the developing polymer chains, resulting in larger aggregates [54,55]. The average size obtained was evaluated using PdI. The Malvern Panalytical Co. Ltd. (Malvern, UK) defines PdI as having dispersibility ranging from 0.05, indicating a high monodispersity, to 0.70, indicating a broad size distribution. Therefore, according to the PdI results in Figure 2a, Py-TW-APS and Py-SDS-APS demonstrated a fair monodispersity (0.12 and 0.19, respectively). An increase in PdI may suggest a greater tendency for particles or molecules to agglomerate [2]. Therefore, the size distribution was significantly larger, presumably due to sample aggregation, such as in the case of Py-CTAB-Fe, PPy*, Py-SDS-Fe, and Py-CTAB-APS.

The difference in the zeta potential between PPy samples polymerized with surfactants and PPy* indicated that Py was polymerized within the micelles of the surfactants because Py monomers were predicted to be located in the hydrophobic core of the surfactant micelles because of the hydrophobic structure of the pyrrole monomer (Log P = 0.75, where P is the partition coefficient between octanol and water). Consequently, the zeta potential of PPy is determined by the type of hydrophilic head of the surfactant facing the exterior [56,57]. Additionally, the high negative charge in Py-SDS-APS and Py-SDS-Fe tended to increase the stability of the sample more than the positive charge due to hydrodynamic interactions and electrostatic repulsion between particles [51]. However, if the size of the sample is large, the zeta potential is affected by gravitation more than hydrodynamic forces [42], such as in the case of Py-SDS-Fe, which shows a negative charge and large size. For Py-CTAB-Fe, PPy* and Py-CTAB-APS contain a positive charge, which makes them simpler to aggregate, corresponding to a high PdI and large average size. However, in the case of Py-TW-APS, this sample also showed a positive charge, as confirmed by the intrinsic properties of the PPy structure. However, TW can stabilize PPy in water because it absorbs onto the surface of particles and provides steric stabilization, where the physical barrier of the absorbed surfactant layer protects particles from coming close enough to aggregate [58]. 

The results from the average size and zeta potential refer to the interaction between the Py monomer, surfactant, and oxidant during polymerization based on the hydrophobic/hydrophilic properties and electrostatic forces among them. These reactions can lead to different morphologies. As shown in Figure 3a, SEM showed a cluster of particles in PPy*, Py-CTAB-APS, Py-SDS-Fe, and Py-CTAB-Fe, corresponding to the average size and PdI detected by DLS, except in the case of Py-SDS-APS, which found a sheet-like form with a large morphology. 

In the absence of a surfactant, PPy* has a particle form, which has been normally observed in many studies [13,28,29]. The morphology of Py-SDS-APS is sheet like, which might be due to the effect of electrostatic interactions between the natural positively charged PPy* and the head group of the surfactants (SO_3_^−^) [24]. This finding was different from that reported by Zhang et al. [27], who reported that in the presence of APS and FeCl_3_, no regular polypyrrole nanostructure was observed. Their findings differ from those of our study, possibly because PPy prepared with SDS remained at low concentrations close to the CMC of SDS and the proportion between each component was not the same. Therefore, it may affect micelle formation when mixed together. However, they believe that the uneven shape differs from that without SDS, resulting from the residual SDS in the PPy structure after drying.

For polymerization with FeCl_3_ (Py-SDS-Fe), we found a cluster of particle-like forms, which is similar to a study using sodium dodecylbenzenesulfonate (DBSNa) as an anionic surfactant [29]. However, unlike in the case of Yussuf et al. [32], who found that using FeCl_3_ as an oxidizing agent showed fibrillar morphology, they used a lower SDS concentration (around 0.02 M) and only performed a 4 h reaction time at room temperature. It is possible that the short polymerization time and the proportion of monomer and surfactant may have affected PPy morphology [24]. However, their work supported that fibrillar morphology showed better conductivity properties than the globular form when they were used as an oxidizing agent. In this work, we also found that using FeCl_3_ as an oxidant in the SDS system resulted in higher conductivity, as shown in Figure 4b, but it showed a globular shape compared to using APS, which represents a sheet-like structure. Additionally, we found ferric element (Fe) in the PPy structure from electron dispersive X-ray spectroscopy (EDS) when FeCl_3_ was used (as shown in Table A1), which might occur because it cannot be completely removed after dialysis. Therefore, the intrinsic properties may have affected the conductivity of PPy more than the shape morphology of these samples. Additionally, regarding the electrostatic force that might have occurred from Fe^+^ between particles, as mentioned above [54,55], this interaction force may disrupt sheet-like forming and increase porosity in the PPy structure [59]. Some works reported that a specific higher porous PPy structure represented a higher surface area and an increase in its electronic properties [60,61].

However, in the case of Py-CTAB-Fe, even though Fe was found in the structure, Py-CTAB-APS had better conductivity. This might be because the morphology of Py-CTAB-APS tended to resemble a string-bead structure, which is similar to the results obtained by Khadem et al. [31], even though they used a lower CTAB concentration and a polymerization period of 4 h at room temperature and under N_2_ atmosphere. Moreover, Zhang X. et al. [27] prepared polymerization at 0–5 °C for 24 h and obtained a ribbon-like form when CTAB was used with APS and obtained sphere-like PPy using a CTAB surfactant and FeCl_3_, similar to Py-CTAB-Fe in the work by Zoromba et al. [14]. Therefore, by employing CTAB, it is possible to alter its morphology into a ribbon or wire form by the process of self-assembly between the positively charged cations of the cationic surfactants and the negatively charged anions of the oxidizing agent APS (S_2_O_8_^2−^) [27,31], but the oxidizing agent has the effect to form a PPy structure. However, the conductivity of PPy polymerized with CTAB was lower than that of PPy polymerized with SDS in either APS or FeCl_3_. Therefore, SDS is a dopant that is useful to dope a conductive structure of PPy better than CTAB. In addition, by predicting the average size, zeta potential from the zetasizer, and morphological structure from SEM, we can estimate the interactions between the Py monomer and surfactant, as shown in Figure 2. 

As shown in Figure 2, the Py monomer aggregation during polymerization is affected by the interaction between surfactants and oxidants, resulting in various morphologies. In addition, their interactions led to distinct chemical structures in their PPy, as determined the chemical structure of PPy samples from Raman spectroscopy, which found varying peak characteristics, as shown in Figure 5a. By studying the chemical structure observed via Raman spectroscopy, we could define whether the morphology is related to the intrinsic chemical structure of PPy. As mentioned in the results section, we found that in Py-TW-APS and Py-CTAB-Fe, there was no peak in the bipolaron structure, which is similar to the peak characteristic of PPy deprotonated by ammonium hydroxide from Trchová et al. [40], which represents a low-oxidative structure. In addition, PPy polymerized with FeCl_3_ showed a sphere-like structure, and Raman spectra characteristics similar to those work (PPy*), even though they used 633 nm or 785 nm laser excitation. In addition, they used dye acids (methyl orange and Acid Blue 25) to dope a bipolaron structure and found that the peak at bipolaron increased as to form a nanotube structure with higher conductivity, which is also similar to other works [62,63]. 

Moreover, as in our previous work, we observed that the ratio between the bipolaron and polaron of PPy polymerized with SDS (sheet-like form) differed from that of PPy without a surfactant (sphere-like form), particularly in the C-H deformation area. Additionally, the bipolaron and polaron structures refer to the oxidative structure of PPy [35,36,37]; a higher bipolaron and polaron ratio indicates higher conductivity properties. Therefore, the ratio between the bipolaron and polaron was calculated from the peak intensity, including the three main areas, as shown in Figure 5b. The graph shows that the C-H deformation area represented the bipolaron/polaron ratio of each sample and showed that PPy polymerized with SDS had a higher ratio of bipolaron and polaron than other conditions, corresponding to the high UV-Vis-NIR absorption at around 960 nm (Figure 6a) and low bandgap value calculated from absorbance (Figure 6b). This phenomenon is consistent with the negatively charged surface and high-conductivity measurements from instruments of Py-SDS-APS and Py-SDS-Fe. Therefore, the correlation between them revealed that SDS incorporated into the PPy structure was the major dopant that enhances the conductivity properties of PPy by interfering with the PPy structure and raising the bipolaron structure in the C–H deformation area, which might be due to the electrostatic interaction between positively charged PPy and the anionic head group of SDS conducting the PPy structure. 

Additionally, Figure 7 represents the conclusion of the relationship between each PPy property in this work. We can categorize them into two groups: low bandgap and high bandgap (red dot line), in which the low bandgap correlated to high bipolaron and polaron ratios (black dot line), which correlated to the theory and other works [50,51]. In this category, it confirmed the relationship between using SDS in Py polymerization and electronic properties (conductivity and bandgap) and bipolaron and polaron ratios in C-H deformation but did not link to the morphology. 

However, the bipolaron and polaron ratio results went against the first hypothesis, which said that the morphology should match the bipolaron and polaron ratio in C-H deformation. However, the fact that the rise in the bipolaron and polaron ratios in the C-H deformation region is linked to SDS provides us with insight into where SDS affects the intrinsic chemical structure of PPy, which might be a criterion leading to higher conductance. The data provide additional underpinnings for the future development of PPy properties.

## 4. Materials and Methods

### 4.1. Materials

Py was used without further purification (98% reagent grade, Sigma-Aldrich Co. LLC, St. Louis, MO, USA). SDS, CTAB, TW, APS, and FeCl_3_ were purchased from Wako Pure Chemical Industries Ltd. (Osaka, Japan). All aqueous solutions required ultrapure water with a resistance of 18.2 Mohm (MΩ) (Direct-Q 3 UV system, Merck Millipore, Osaka, Japan). Indium tin oxide (ITO) glass electrodes were purchased from BAS Inc., Tokyo, Japan. Silver paint with a resistance less than or equal to 1 Ω was purchased from Polycalm, a company based in Japan. A dialysis tube with a molecular weight cutoff of 14 kDa was purchased from NaRiKa Corporation, Tokyo, Japan.

### 4.2. PPy Synthesis

PPy samples were synthesized via chemical polymerization. Py was initially introduced into a surfactant solution at a concentration of 0.10 M, which may be either SDS, CTAB, or TW. Subsequently, an oxidizing agent solution (APS or FeCl_3_) was slowly added to the mixture during agitation to achieve a concentration of 0.10 M (the oxidizer and monomer ratio was 1). The mixture was agitated for 24 h at room temperature (25 °C), resulting in the formation of a PPy suspension with a dark color. The synthesized PPy was subjected to dialysis in ultrapure water to eliminate residual impurities. The PPy sample was freeze-dried and stored in a desiccator until use.

### 4.3. Particle Size Distribution and Zeta Potential Measurement

PPy was suspended in ultrapure water (0.10–0.20 mg/mL). The suspension was tested for particle size distribution and zeta potential using a zetasizer apparatus (ZEN5600, Zetasizer, Malvern Instruments Ltd., Worcestershire, UK). The experiment was performed at 25 °C.

### 4.4. Morphology Observation by Scanning Electron Microscopy (SEM) 

The dried PPy powder was obtained from the freeze dryer and stored in a desiccator until analysis. The PPy powder was adhered to an electrically grounded sample holder using a double-face conducting tape. The morphology of the dried PPy powder was analyzed and manipulated using scanning electron microscopy (JCM-7000, JEOL Ltd., Tokyo, Japan). The pictures from SEM were evaluated on the edges and shape of particles using Image J software (ImageJ Version 1.53 K, National Institutes of Health, MD, USA).

### 4.5. Electrical Conductivity Analysis

The resistance of the PPy powder was determined using a chemiresistive device, as shown in Figure 3 [24]. Briefly, PPy powder (0.005–0.008 g/cm^2^) was cast into pellets using a hand press and sandwiched between two ITO glass electrodes. The distance between the electrodes and the thickness of the pellet were maintained at their initial values using 0.80 mm thick silicone rubber. The resistance of PPy was monitored and recorded based on the two-point probe measurement principle operated by an automatic polarization system (HZ-7000, Hokuto Denko Corporation, Tokyo, Japan). Each resistance value was obtained from the slope of the *I*/*V* plot based on Ohm’s law (*V* = *IR*). From the *I*/*V* plot, *R* is the reverse of slope (*R* = 1/slope), which is determined by Origin software (OriginPro 2023, Student version 10.0.0.154, OriginLab Corporation, Northampton, MA, USA). The conductivity of PPy was determined by the correlations shown in Equations (1) and (2) as follows: (1)$$\rho =\frac{RA}{L}$$
(2)$$\rho =\;\frac{1}{\sigma }$$
where *R* is the resistance of PPy (Ω), *ρ* is the resistivity of PPy (Ω·m), L is the thickness of the PPy pellet (8.00 × 10^−4^ m), A is the area of the sample (1.96 × 10^−5^ m^2^), and σ is the electrical conductivity (S/m). 

### 4.6. Raman Spectroscopy

Raman spectra of PPy were analyzed using Raman spectroscopy (HR-800, Horiba Ltd., Kyoto, Japan) with a laser excitation wavelength of 532 nm. The peak intensities of the bipolarons and polarons in PPy were evaluated using Origin software (Version 2023, OriginLab Corporation, Northampton, MA, USA). Subsequently, the ratio between the highest intensity of the bipolarons and polarons was analyzed.

### 4.7. Ultraviolet/Visible/Near Infrared (UV-Vis-NIR) Absorbance and Bandgap Calculations

PPy was diluted and suspended in ultrapure water (0.10–0.20 mg/mL) in a quartz cuvette with a path length of 10 mm, and its optical characteristics were investigated using UV-Vis-NIR spectroscopy (UV1800, SHIMADZU, Kyoto, Japan). The optical bandgap was calculated using Tauc’s correlation, as indicated in Equation (3).
(3)$${\left(\alpha hv\right)}^{1/n}=\;\alpha_{o}(hv-E_{g})$$
where *α* is the absorption coefficient, *hv* is the photon energy (eV), *h* is Planck’s constant (6.6261 × 10^−34^ J·s), and *ν* is the photon frequency with *ν* = *C/λ,* where *C* = the speed of light (2.998 × 10^8^ m/s), *λ* is the wavelength (nm), and the photon energy (eV) is 1240/*λ*. *E_g_* is the optical bandgap (eV), *α_o_* is the constant band-tailing parameter, and *n* is the power factor (*n* = 2 for an indirect transition bandgap) [51]. The plot of *(αhν)*^1/n^ versus photon energy (*hν*) will show a straight line inside a given region. The optical bandgap was calculated by extending the straight line intercept along the (*hν*)-axis.

## 5. Conclusions

Compared to Py-SDS-APS, Py-CTAB-APS, and Py-TW-APS, the morphology shifts towards a sheet-like configuration when using SDS as a surfactant and shows a higher bipolaron and polaron ratio in the C–H deformation and higher electronic properties. In contrast, Py-SDS-Fe showed a cluster of particle-like form, and the conductivity and the bipolaron and polaron ratio in the C–H deformation of Py-SDS-Fe surpassed that of Py-CTAB-Fe. The results showed that if different oxidants were used, the shape morphology of PPy might not have an effect on the bipolaron and polaron ratio in C–H deformation. However, the ratio of bipolaron to polaron has a strong relation with conductivity. This work highlights that SDS contributes to elevating the bipolaron content within the C–H deformation region of PPy, which may occur through the electrostatic interaction between PPy and the head group of SDS, resulting in the enhancement of its conductivity.

## Data Availability

Data are contained within the article.

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
