# Peer review of "Exploring the Influence of Morphology on Bipolaron–Polaron Ratios and Conductivity in Polypyrrole in the Presence of Surfactants"

_molecules, 2024, doi:10.3390/molecules29061197_

Round 1

Reviewer 1 Report

Comments and Suggestions for Authors

Very interesting work on bipolaron to polaron ratio in the PPy synthesized with different surfactans. I have few suggestions to improve the quality of the manuscript even further:

1. Authors should enlarge (while maintaining the quality of the image) Figures 2 and 5. Especially in terms of SEM observations, it is necessary to provide well defined images, because, even at 175% magnification, the quality of the figure is low.

2. I would suggest performing XPS measurements of the PPy samples and including the detailed N1s deconvoluted spectrum analysis, as it could provide additional information on the nitrogen species in polaron and non polaron states.

3. In the introduction, to improve clarity for the readers, I would suggest to implement/introduce a scheme/figure with general concept of the pyrrole monomer polymerization pathway.

Author Response

Response to Reviewer I Comments

1. Summary

Thank you very much for your thorough review of our manuscript. Below, you will find detailed responses to each of your comments. All changes and corrections have been highlighted in blue and marked as comments for easy identification in the revised manuscript we have resubmitted.

2. Questions for General Evaluation

Reviewer’s Evaluation

Response and Revisions

Does the introduction provide sufficient background and include all relevant references?

Yes

-

Are all the cited references relevant to the research?

Yes

-

Is the research design appropriate?

Yes

-

Are the methods adequately described?

Yes

-

Are the results clearly presented?

Can be improved

Please consider the response as below

Are the conclusions supported by the results?

Yes

-

3. Point-by-point Response to Comments and Suggestions for Authors

Comments and Suggestions for Authors: Very interesting work on bipolaron to polaron ratio in the PPy synthesized with different surfactans. I have few suggestions to improve the quality of the manuscript even further:

Comments 1: Authors should enlarge (while maintaining the quality of the image) Figures 2 and 5. Especially in terms of SEM observations, it is necessary to provide well defined images, because, even at 175% magnification, the quality of the figure is low.

Response 1: I appreciate your consideration on this issue. We agree with this statement. Therefore, we have expanded Figure 3 and Figure 6, which are shown on Page 5, Lines 184–192, and Page 8, Line 274 in the revised manuscript, respectively.

Comments 2: I would suggest performing XPS measurements of the PPy samples and including the detailed N1s deconvoluted spectrum analysis, as it could provide additional information on the nitrogen species in polaron and non polaron states.

Response 2: We appreciate your suggestion of using XPS data to examine polypyrrole (PPy) samples. XPS analysis gives bipolaron and polaron data by evaluating the nitrogen species states.

However, after careful consideration, Raman spectroscopy's exceptional ability to study and characterize PPy bipolarons and polarons was the driving force behind the decision. The inquiry includes the assessment of C-C ring stretching, C-H ring deformation, and C-C ring deformation. This approach contributes to our study's objectives by exposing the electrical and structural alterations of PPy under various surfactants. The C-H deformation zone for each sample was chosen based on its relevance to the study's objectives and agreement with our research issue.

Logistic restrictions also affected our strategy. We cannot do XPS analysis because of a lack of XPS equipment. We see XPS's potential to supplement our findings and want to examine this topic in future research if resources allow.

I would appreciate your input again. It is critical to broaden our research perspectives and advance our science.

Comments 3: In the introduction, to improve clarity for the readers, I would suggest to implement/introduce a scheme/figure with general concept of the pyrrole monomer polymerization pathway.

Response 3: I appreciate your observation. We agree with the stated statement. Therefore, Figure 1 illustrating the pyrrole polymerization has been inserted on Page 3, Lines 104–119, and referenced on Page 2, Line 76, and Line 84.

4. Response to Comments on the Quality of English Language

Point 1: English language fine. No issues detected.

Response 1: Thank you for your evaluation.

Reviewer 2 Report

Comments and Suggestions for Authors

The manuscript submitted by Samwang and coworkers aims to contribute to the field of PPy synthesis and its properties, trying to elucidate several key factors that affect morphology, bipolaron/polaron ratios, and electroactivity/conductivity depending on the use (and the kind) of surfactant. The study is easy to read and the conclusions are supported by different experimental data, comparing them with other works available in the literature. I am attaching some comments/suggestions that could be useful to authors in order to improve the presented manuscript.

1- Introduction.

 Some references are missing. Please, check lines 33-36, and 52-55.

 Line 72, considering the APS acronym was previously defined.

 Line 95, please clarify the spectra type (Raman?).

2- Results.

Figure 1a. Y axis: PDI maximum value has to be 1 for DLS.

Line 161, is needed to add a reference about the melting point of Tween80.

Line 268. Why did the authors refer to the drug delivery system particularly?

Line 323. Please, show EDS results if possible. 

3- Materials and Methods.

Line 410. Please, check the resistance of ultrapure water.

Line 421. Please, give details about the dialysis membrane cutoff.

4- Conclusions.

I recommend a re-structuration of this section.

Author Response

Response to Reviewer II Comments

1. Summary

Thank you very much for your thorough review of our manuscript. Below, you will find detailed responses to each of your comments. All changes and corrections have been highlighted in yellow and marked as comments for easy identification in the revised manuscript we have resubmitted.

2. Questions for General Evaluation

Reviewer’s Evaluation

Response and Revisions

Does the introduction provide sufficient background and include all relevant references?

Can be improved

Please consider the response as below

Are all the cited references relevant to the research?

Can be improved

Please consider the response as below

Is the research design appropriate?

Yes

-

Are the methods adequately described?

Yes

-

Are the results clearly presented?

Yes

-

Are the conclusions supported by the results?

Yes

-

3. Point-by-point response to Comments and Suggestions for Authors

Comments and Suggestions for Authors: The manuscript submitted by Samwang and coworkers aims to contribute to the field of PPy synthesis and its properties, trying to elucidate several key factors that affect morphology, bipolaron/polaron ratios, and electroactivity/conductivity depending on the use (and the kind) of surfactant. The study is easy to read and the conclusions are supported by different experimental data, comparing them with other works available in the literature. I am attaching some comments/suggestions that could be useful to authors in order to improve the presented manuscript.

Comments 1: Introduction.

(1.1) Some references are missing. Please, check lines 33-36, and 52-55.

(1.2) Line 72, considering the APS acronym was previously defined.

(1.3) Line 95, please clarify the spectra type (Raman?).

Response 1: Thank you for pointing this out. Therefore, we have made alterations to some particulars and highlighted them in yellow, as seen in the revised manuscript:

(1.1)  On Page 1, Lines 33–37, and Page 2, Lines 53–56, we have added references, which give information about the properties and chemical structure of polypyrrole during polymerization and about the soft template, respectively. These additions aim to support the details in that paragraph.

(1.2)  Page2, Line 72, the full name of ammonium persulfate was removed, and the abbreviation “APS” was used instead.

(1.3)  Page2, Line 95, we have added a sentence that is “from the Raman spectra” to clarify the type of spectra.

Comments 2: Results.

(2.1)  Figure 1a. Y axis: PDI maximum value has to be 1 for DLS.

(2.2)  Line 161, is needed to add a reference about the melting point of Tween80.

(2.3)  Line 268. Why did the authors refer to the drug delivery system particularly?

(2.4)  Line 323. Please, show EDS results if possible.

Response 2: We appreciate your feedback. We agree with these remarks; therefore, we have modified certain particulars and emphasized them in yellow to indicate the areas where we made revisions, as shown in the revised manuscript:

(2.1)  Page 5, Line 163, the maximum value of PdI has changed to be 1 as shown in Figure 1a.

(2.2)  Page 5, Line 177-178, the author provided the source referring to the melting point value of Tween 80 (TW) obtained from a Safety Data Sheet of TCI America.

(2.3)  On Page 8, Line 293-294, To enhance reader comprehension, I have revised the context surrounding PdI. Specifically, I have eliminated the context about the drug delivery system and instead cited the reference from Malvern Panalytical Co. Ltd., the equipment brand employed in this study.

(2.4)  The EDS result was referred to on Page 9, Line 348 in the revised manuscript and provided the data in the table in Appendix A Section on Page 14, Line 527-528.

Comments 3: Materials and Methods.

(3.1) Line 410. Please, check the resistance of ultrapure water.

(3.2) Line 421. Please, give details about the dialysis membrane cutoff.

Response 3:  We appreciate your point of view. The material has been corrected and included in the revised manuscript.:

(3.1)  Page 12, Line 435, the unit of resistance was updated to be 18.2 Mohm (MΩ).

(3.2)  Page 12, Line 438-439, the information of dialysis tube was updated in Section 4.1. Materials.

Comments 4: Conclusions.

 I recommend a re-structuration of this section.

Response 4: Thank you for your point. We agree with this suggestion, therefore, we have updated the format of the conclusions section into one paragraph, as shown on Page 13, Line 502-513.

4. Response to Comments on the Quality of English Language

Point 1: English language fine. No issues detected

Response 1: Thank you for your evaluation.
